# Revised methodology for CO$_2$ and CH$_4$ measurements at remote sites using a working standard gas saving system

Motoki Sasakawa[1], Noritsugu Tsuda[2], Toshinobu Machida[1], Mikhail Arshinov[3], Denis Davydov[3], Aleksandr Fofonov[3], Boris Belan[3]

[1]Center for Global Environmental Research, Earth System Division, National Institute for Environmental Studies, Tsukuba, 305-8506, Japan
[2]Global Environmental Forum, Tsukuba, 305-0061, Japan
[3]V.E. Zuev Institute of Atmospheric Optics, Russian Academy of Sciences, Siberian Branch, Tomsk, Russia

*Correspondence to:* Motoki Sasakawa (sasakawa.motoki@nies.go.jp)

**Abstract.** We have revised a calculation method of mole fractions and uncertainties for in-situ CO$_2$ and CH$_4$ measurements with a working standard gas saving system. It uses on-site compressed air to track the baseline drift of sensors. Japan-Russia Siberian Tall Tower Inland Observation Network (JR-STATION) was made up of this system, which was installed across nine different sites in Siberia. The system acquires semi-continuous data by alternating between sampling air from multiple altitudes through switched flow paths and recording several minutes of averaged data for each altitude. We estimated the sensor

repeatability ($u_r$) based on the measurement of on-site compressed air. The $u_r$ for CO$_2$ and CH$_4$ mostly ranged around 0.05 ppm and below five ppb, respectively. The combined standard uncertainties ($u_c(x)$) of time-averaged ambient air measurements were sometimes higher than the $u_r$ for each period because the data included atmospheric variability during the measurement period of several minutes. Data users should consider the difference between the $u_r$ and $u_c(x)$ to select optimal data, depending on their focusing spatial scale. The CO$_2$ and CH$_4$ data measured with a non-dispersive infrared (NDIR)

analyzer and a tin dioxide sensor (TOS) exhibited good agreement with those measured by a cavity ring-down spectroscopy (CRDS).

## 1 Introduction

It is known that accurate measurements of greenhouse gas mole fractions require the analyzers to be calibrated against a set of standard gas mixtures. At least one of them (target) should be used hourly to track an NDIR analyzer's baseline drift (Andrews

et al., 2014). Delivering high-pressure cylinders to remote sites is a significant issue for long-term atmospheric monitoring. Thus, to reduce the consumption of gas, Watai et al. (2010) developed a system that utilizes on-site air as sub-working standard gas (SWS-gas) to track the baseline drift of the NDIR sensors. Watai et al. (2010) then installed this system at a remote tower site at Berezorechka (56°08′45″N 84°19′55″E) in West Siberia in 2001 to measure CO$_2$ mole fractions semi-continuously. After this, in Central Siberia, Winderlich et al. (2010) developed a measuring system without dehumidification using a CRDS

analyzer to reduce the frequency of cylinder replacement. The CRDS is a more stable device, and a calibration frequency of every two weeks to every month is recommended (ICOS RI, 2020).

   Concerning $CH_4$ measurement, a commonly used gas chromatograph with a flame ionization detector (GC/FID) requires hydrogen and carrier gases. It also needs significant power consumption. However, electric power is often restricted at remote sites. Thus, Suto and Inoue (2010) modified a tin dioxide sensor (TOS), which is widely used to detect natural gas leaks, to be

able to measure $CH_4$ in the atmosphere. The developed TOS unit does not need hydrogen and carrier gases. The nominal power consumption for the unit, consisting of TOS, temperature-stabilizer mechanism, and electronic circuits for the sensor and heater, is less than 10W.

   We added the TOS unit to the system at the tower site in West Siberia, then expanded the tower observation network (Sasakawa et al., 2010; Sasakawa et al., 2012; Sasakawa et al., 2013). The tower network named JR-STATION now consists of six tower

sites in West Siberia. Recently, we added CRDS analyzers at Karasevoe (58°14′44″N 82°25′28″E) in 2015 (Picarro G2401), and at Demyanskoe (59°47′29″N 70°52′16″E) and Noyabrsk (63°25′45″N 75°46′48″E) in 2016 (Picarro G2301) to improve the robustness of the measurements.

   The system follows the operational method conceived under the instrumentation constraints a quarter of a century ago (around 2000) and in remote areas with limited infrastructure (Watai et al., 2010). However, Watai et al. (2010) did not present an

optimal calculation method for the measurement sequence of this system (nor did the TOS calculation method). Nor did they calculate uncertainties, especially as has been recommended in the GAW report (2020) in recent years.

   Thus, we have updated the calculation method for calculating $CO_2$ and $CH_4$ mole fractions to derive their uncertainty for each data set simultaneously. Here, we describe the details of the modified measurement system and the calculation method. Furthermore, some sites have installed CRDS, which have been widely used for greenhouse gas observations (Kwok et al.,

2015) and allow partial comparisons with conventional sensors, so the recalculated data were compared with the CRDS data to see how well they agree.

## 2 Method

### 2.1 Measurement system

   Ambient air was taken from air sample inlets at two different heights (four at Berezorechka) on television and radio-relay

communication towers (Table 1). Each sample inlet was mounted several meters away from the tower at the end of an extension arm. The air from the inlets was pulled into the measurement system through the sampling lines (6-mm OD Decabon tube). The measurement system was housed in a freight container insulated to reduce temperature variation. Two thermometers were mounted inside the container, one near the ceiling and the other near the floor. According to the upper thermometer, the room temperature in the container during the year was kept above 15ºC and the temperature difference in the 12-hour calibration

interval was kept below 3ºC on average during the year. Since the introduction of the CRDS, a simple cooler was installed to prevent the temperature inside the container from rising too high during the summer months due to the heat generated by the

CRDS. A schematic diagram of the measurement system is shown in Fig. 1. The measurement system consists of a pump unit, a selector unit, and an analyzer unit. The pump unit was located upstream of the selector and analyzer unit to keep the downstream pressure higher than the ambient, which reduced the likelihood of bias in measurements due to any leak from many connections in the system. Two diaphragm pumps (model N86KNE, KNF, Germany) delivered the sample air into the system. The sampling lines were flushed continuously with a flow rate of about seven standard liters per minute, and excess air was vented through the back-pressure valve ("BPV" in the pump unit). Then the air was dried by an adiabatic expansion in a glass water trap ("WT" in the pump unit) that was purged every hour via an NC solenoid valve, which was opened twice for 10 seconds to remove the condensed water. The sample air was also dried using a semipermeable membrane dryer (PD-625–24SS, Permapure, USA) ("Nafion" in the selector unit). The semipermeable membrane dryer removed water vapor from the pressurized inner tube to an outer tube where the split gas flowed (split sample method). The air from the upper and lower-level inlet, the three working standard gases (WS-gases), and the sub–working standard gas (SWS-gas) were selected through a 6-port valve manifold. While the WS-gases or the SWS-gas flowed into the analyzer unit, the sample air was exhausted at the 6-port valve. In the analyzer unit, the sampled air was extra dried with magnesium perchlorate, which was fed into a stainless steel tube with a dimension of 2 cm in inner diameter and 10 cm in length ("$Mg(ClO_4)_2$" in the analyzer unit). There were two tubes, and the flow path of the air switched from one to the other every month. The used magnesium perchlorate was replaced before the next run. After being dried with the magnesium perchlorate, the air retained its dewpoint at around -50 ºC (39 ppm). The dehumidified air was then introduced into an NDIR analyzer (LI-820, LI-COR, USA; LI-7000 was used until September 2008 at BRZ) at a constant flow rate of 35 standard cubic centimeters per minute (sccm) set by a mass flow controller (SEC-E40, STEC, Japan). The $CO_2$ mole fraction was defined as the mole fraction in the dry air, and water vapor correction was not adopted. After passing through the NDIR, the air flowed into the TOS unit. A chemical desiccant made of $P_2O_5$ was installed in front of the TOS because it is necessary to keep water vapor below ten ppm in the sample air for this type of sensor. The sensor was designed to operate in areas lacking the sufficient infrastructure to sustain a conventional measurement system, such as a significant power source, carrier gas supply, and temperature-stabilized environment. The sensor has been verified against a gas chromatograph equipped with a flame ionization detector (Suto and Inoue, 2010). We additionally installed the CRDS (Picarro Inc.) analyzer at Karasevoe (G2401) in 2015, and at Demyanskoe and Noyabrsk (G2301) in 2016 to improve the system. The sampled air was split after leaving the 6-port valve, then fed into the CRDS at a constant flow rate of 35 sccm set by a mass flow controller (SEC-E40, STEC, Japan) through a semipermeable membrane dryer (model MD-050-72S-1, Permapure, USA). To protect the cavity of the CRDS from an inflow of the dissolved chemical desiccant ($Mg(ClO_4)_2$ or $P_2O_5$) in the accidental case of a broken pump etc., we equipped the CRDS with two poppet check valves ("PCV" in the analyzer unit). When the pumps in the pump unit stop and only the CRDS pump is running, the flow stops at the PCV upstream of $P_2O_5$, and the increased suction pressure allows air in the container to enter from the PCV in front of Nafion. The data from this process has been deleted.

Three WS-gases (STD1, STD2, STD3) were prepared from pure $CO_2$ and $CH_4$ (G1 grade, Japan Fine Products Corp. (JFP), Japan) diluted with purified air (G1 grade, JFP), and their mole fractions were determined against the NIES 09 $CO_2$ scale

(Machida et al., 2011) and NIES 94 $CH_4$ scale. Each scale was established by a series of standard gases prepared by the gravimetric method. Since the pure $CO_2$ gas is derived from burned petroleum, the isotopic $CO_2$ composition of the gases shows lighter than atmospheric $CO_2$. When the NDIR analyzer is calibrated against $CO_2$ standards with lighter-than-atmospheric $CO_2$ isotopic composition, the NDIR analyzer measures a lower $CO_2$ mole fraction in a sample air with a known
$CO_2$ mole fraction. The error in the apparent NDIR $CO_2$ mole fraction depends on its individual sensitivity to the optical filter property. Tohjima et al. (2009) reported that the errors for the three NDIR analyzers range from -0.04 to -0.08 ppm. Compared to the WMO-CO2-X2007 scale, the NIES 09 $CO_2$ scale is consistent within 0.1 ppm (Round Robin 5 and 6 Comparison Experiment). Since there have yet to be published results with WMO-CO2-X2019, scale conversion could be the linear function shown in Hall et al. (2021) ($X2019 = 1.00079 \times NIES09 - 0.142$ (ppm)). The NIES 94 $CH_4$ scale ranges from 3.0 to
5.5 ppb higher than the WMO-CH4-X2004A scale (Round Robin 5 and 6 Comparison Experiment).

*Table 1*. Main features of the towers in the network of tall towers used for continuous long-term atmospheric $CO_2$ and $CH_4$ measurements over Siberia.

| Identifying Code | Location | Latitude | Longitude | Air inlet heights (m) | Elevation at tower base (m a.s.l)[1] |
|---|---|---|---|---|---|
| BRZ | Berezorechka | 56°08′45″ | 84°19′55″ | 5, 20, 40, 80 | 168 |
| KRS | Karasevoe | 58°14′44″ | 82°25′28″ | 35, 67 | 76 |
| IGR | Igrim | 63°11′30″ | 64°24′50″ | 24, 47 | 9 |
| NOY | Noyabrsk | 63°25′45″ | 75°46′48″ | 21, 43 | 108 |
| DEM | Demyanskoe | 59°47′29″ | 70°52′16″ | 45, 63 | 63 |
| SVV | Savvushka | 51°19′31″ | 82°07′42″ | 27, 52 | 495 |
| AZV | Azovo | 54°42′18″ | 73°01′45″ | 29, 50 | 110 |
| VGN | Vaganovo | 54°29′50″ | 62°19′29″ | 42, 85 | 192 |
| YAK | Yakutsk | 62°05′19″ | 129°21′21″ | 11, 77 | 264 |

[1]Approximate estimates from Google earth.


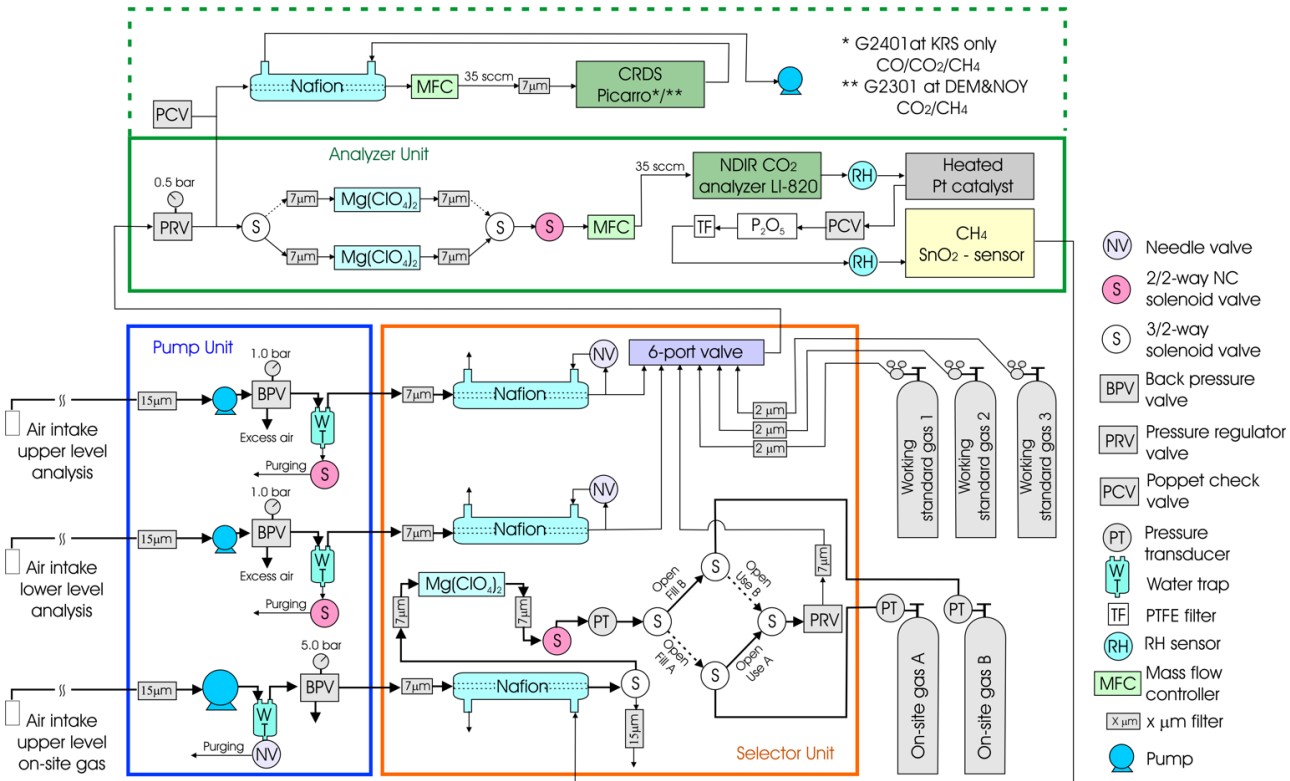

**Figure 1. Schematic diagram of tower observation system.**

A frequent calibration with WS-gases within 1-2 hours is necessary to conduct precise measurements of $CO_2$ and $CH_4$ because the output of the NDIR analyzer or the TOS could vary depending on the environment (atmospheric pressure etc.) in 1-2 hours. But if the calibration were done at this frequency, standard gases would be consumed in less than a year. Because delivering WS-gases to remote sites is a significant issue, we utilized on-site compressed air as SWS-gas to track the sensors' baseline drift, which reduced the consumption of the three standard gases. The on-site compressed air ("On-site gas A/B" in Fig. 1) was analyzed every hour, and the WS-gases were measured every 12 hours to calibrate the sensors (details of the sequence are shown below). The measurement protocol adhered to the procedure established by Watai et al. (2010) for NDIR analysis using this system. However, at sites where the CRDS was installed, the WS-gas measurement interval was extended to 48 hours to prolong the longevity of the WS-gas. An aluminum cylinder (0.048 m³) for SWS-gas was automatically exchanged when the inner pressure decreased below 0.1 MPa, then soon air from the highest inlet was compressed by a pump (LOA-P103-NO, GAST, USA) into the cylinder for about 5 hours to approximately 0.35 MPa, after having been passed through a similar triple dehumidification path as the sampled air (a stainless steel water trap, a semipermeable membrane dryer (SWF- M06–400, AGC, Japan), and magnesium perchlorate). It was preserved for approximately one week (three days with the CRDS) for usage until the inner pressure in one used for measurements decreased below 0.1 MPa (Table 2). Schibig et al. (2018) reported that

the $CO_2$ mole fraction in a 29.5 L aluminum cylinder increases by $0.090 \pm 0.009$ μmol mol$^{-1}$ when dropping from 150 bar to 1 bar, but also note that this change is smaller if larger cylinders are used. Given the cylinder size and filling pressure in this

system, mole fraction changes within the cylinder are considered negligible. The variations in SWS mole fraction with CRDS between WS-gases (48 hours) were in fact very stable regardless of the SWS mole fraction range (Fig. S1-2).

Air temperature and relative humidity were measured at both heights on the tower using commercial sensors (HMP45D, Vaisala, Finland). A wind monitor (model 81000, R. M. Young, USA) determined wind direction and speed at the higher inlet. Solar radiation was measured by a pyranometer (CM3, Kipp & Zonen, Netherlands), and precipitation by a tipping bucket rain

gauge (model 52202, R. M. Young, USA) on the top of the container laboratory. The analysis operation and data logging were performed by a measurement and control system (CR10X datalogger, CAMPBELL, USA). Stored data were retrieved once a month when both a system check and replacement of consumables (e.g., chemical desiccants) took place.

*Table 2.* SWS-gas A/B measurement and filling sequence.

| Approximate elapsed time (h) | Trigger (inner pressure) | 3-way solenoid valve in Figure 1 | Cylinder A | Cylinder B |
|---|---|---|---|---|
| 0 | <0.1 Mpa (B) | Solid line | starts flowing | starts being compressed |
| 5 | >0.35MPa (B) | - | - | stop compression |
| 168 | <0.1 Mpa (A) | Dash line | starts being compressed | starts flowing |
| 173 | >0.35MPa (A) | - | stop compression | - |
| 336 | <0.1 Mpa (B) | Solid line | starts flowing | starts being compressed |


## 2.2 Measurement sequence

To be able to measure air at two heights, the air-sampling flow path was rotated every 20 min with the 6-port valve in the selector unit; that is, the higher inlet was sampled from hh:00 to hh:20, the lower inlet from hh:20 to hh:40, and the SWS-gas

from hh:40 to (hh+1):00. During the first 17 min of each 20-min sampling interval, the system is flushing to equilibrate the air sample after switching. The final three-minute readouts were averaged and reported as the representative output data for the applicable one-hour period. Measurement frequency was 3 sec; thus, only the average and standard deviation (SD) of 60 readouts in voltage were stored in the CR10X. This was to minimize the data size for the limited storage capacity. The timestamp was the end time of every 20-min measurement interval. The raw data collected with the CRDS analyzer were

stored in the CRDS' hard disk and processed after downloading in our laboratory.

Figure 2 shows the schematic measurement sequence for half a day. In the Fig. 2, we defined when the SWS-gas was measured just before an arbitrary series of WS-gas measurements as $t_0$. Then we numbered the time of the following measurements in turn. We also defined the series of standard gas measurements at the beginning of the 12 hours as "B" and at the end as "E".

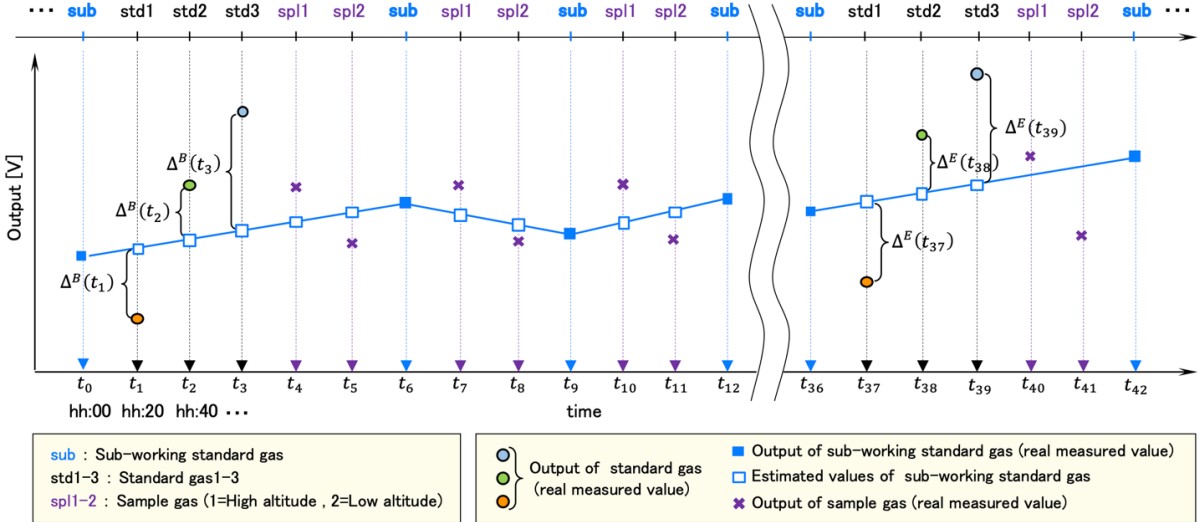

**Figure 2. Measurement sequence for a half day between subsequent measurements of WS-gases.**

## 2.3 Quality check of the standard gas measurements

First, we checked the relationship among three standard gas measurements. We calculated the differences ($\Delta^B(t_i)$, $\Delta^E(t_j)$) between the measured output voltages of the standard gases ($V_{std}(t_i)$, $V_{std}(t_j)$) and the estimated one of the SWS-gas at the time of the standard gas measurement (Fig. 2). Here $i = 1, 2, 3$, and $j = 37, 38, 39$. The output value of the SWS-gas was interpolated by time using the closest output of the SWS-gas before and after the series of standard gas measurements. Thus, these values and their variances are expressed as:

$$\Delta^B(t_i) = V_{std}(t_i) - \left( \frac{6-i}{6} \cdot V_{sub}(t_0) + \frac{i}{6} \cdot V_{sub}(t_6) \right) \tag{1}$$

$$(\sigma^B(t_i))^2 = \left(\sigma_{std}(t_i)\right)^2 + \left(\frac{6-i}{6} \cdot \sigma_{sub}(t_0)\right)^2 + \left(\frac{i}{6} \cdot \sigma_{sub}(t_6)\right)^2 \tag{2}$$

$$\Delta^E(t_j) = V_{std}(t_j) - \left( \frac{42-j}{6} \cdot V_{sub}(t_{36}) + \frac{j-36}{6} \cdot V_{sub}(t_{42}) \right) \tag{3}$$

$$\left(\sigma^E(t_j)\right)^2 = \left(\sigma_{std}(t_j)\right)^2 + \left(\frac{42-j}{6} \cdot \sigma_{sub}(t_{36})\right)^2 + \left(\frac{j-36}{6} \cdot \sigma_{sub}(t_{42})\right)^2 \tag{4}$$

We estimated the output of STD1 at $t_2$ by adding $\Delta^B(t_1)$ to the estimated one of the SWS-gas at $t_2$. We also evaluated the output of STD3 at $t_2$ by adding $\Delta^B(t_3)$ to the estimated one of the SWS-gas at $t_2$. The same estimation was done at $t_{38}$. We then made

a linear calibration line with the output of STD2 and the estimated outputs of STD1 and STD3. Only those sets of the three
standard gas measurements whose coefficients of determination were higher than 0.999 for $CO_2$ and 0.99 for $CH_4$ were adopted
for the following calculation.

The difference in output (voltage) between $\Delta^B$ and $\Delta^E$ for each standard gas was defined as follows:

$$\delta_{(1,37)} = \Delta^E(t_{37}) - \Delta^B(t_1) \tag{5}$$

$$\delta_{(2,38)} = \Delta^E(t_{38}) - \Delta^B(t_2) \tag{6}$$

$$\delta_{(3,39)} = \Delta^E(t_{39}) - \Delta^B(t_3) \tag{7}$$

The $\delta$ must be small unless the system is unstable, e.g., when the sensitivity of the sensors changes considerably for some
reason. To exclude the data obtained during system malfunction, we determined a threshold for $\delta$ by converting it into mole
fraction (<5.0 ppm for $CO_2$, <50 ppb for $CH_4$). Data showing values over the threshold were excluded from the calculation.

The difference in $CO_2$ mole fraction between sides $B$ and $E$ was calculated as follows:

$$\overline{\delta_{(i,j)}^B} = \left| \delta_{(i,j)} / S^B \right| \tag{8}$$

$$\overline{\delta_{(i,j)}^E} = \left| \delta_{(i,j)} / S^E \right| \tag{9}$$

where $S^B$ and $S^E$ are the slopes of the linear regression line at sides $B$ and $E$. Because the x-axis of the calibration line for $CH_4$
is the logarithm of the mole fraction, the difference in $CH_4$ mole fractions was calculated as:

$$\overline{\delta_{(i,j)}^B} = C_i \cdot \left| e^{\frac{\delta_{(i,j)}}{S^B}} - 1 \right| \tag{10}$$

$$\overline{\delta_{(i,j)}^E} = C_i \cdot \left| e^{-\frac{\delta_{(i,j)}}{S^E}} - 1 \right| \tag{11}$$

where $C_i$ is the mole fraction of the standard gas.

## 2.4 Calculation of the sample mole fraction and the combined standard uncertainty

The analysis precision for this system under laboratory conditions was uniformly estimated as 0.3 ppm for $CO_2$ (Watai et al.,
2010). Concerning $CH_4$ precision, Sasakawa et al. (2010) estimated it as 3.0 ppb based on the result of Suto and Inoue (2010).
However, the experiment condition by Suto and Inoue (2010) was different from the gas saving system. Instead, they connected
only the WS-gases to the TOS, then reported the SD of repeated measurements. The $CH_4$ analysis precision for this system
thus could be more significant than 3.0 ppb. Furthermore, the sensitivity and stability of the sensor could differ depending on
the individual sensor and the condition of the individual system. We thus have updated the method for calculating the $CO_2$ and
$CH_4$ mole fractions to derive their combined standard uncertainty for each data simultaneously.

### 2.4.1 Estimation of the output values of working standard gases and their SD at the time of the air sample measurements

We estimated the outputs in voltage of three standard gases at each measurement time of the sample air by interpolating the outputs of the three WS-gases depending on the difference of the outputs in voltage of the SWS-gas only when both standard gas measurements satisfied the criteria described in section 2.3. Depending on the time difference between the targeted sample ($t_k$; $k = 4,5,7,8, \ldots 34, 35$) and standard gases at both sides $B$ and $E$ (($t_i$, $t_j$); ($i, j$) = (1, 37), (2, 38), (3, 39)), the representative value ($\hat{V}^{BE}_{std(t_i,t_j)}(t_k)$) and its variance ($\left( \hat{\sigma}^{BE}_{std(t_i,t_j)}(t_k) \right)^2$) were estimated as follows (Fig. 3):

Figure 3. Schematic diagram of the estimation method for the output of the standard gas at the time of sample air measurement.

$$\hat{V}^{BE}_{std(t_i,t_j)}(t_k) = \frac{a}{a+b} \cdot \hat{V}^{B}_{std(t_i)}(t_k) + \frac{b}{a+b} \cdot \hat{V}^{E}_{std(t_j)}(t_k)$$

$$= \hat{V}_{sub}(t_k) + \frac{1}{a+b} \cdot \left\{ a \cdot \Delta^{B}(t_i) + b \cdot \Delta^{E}(t_j) \right\} \tag{12}$$

$$\left( \hat{\sigma}^{BE}_{std(t_i,t_j)}(t_k) \right)^2 = \left( \hat{\sigma}_{sub}(t_k) \right)^2 + \left( \frac{a}{a+b} \cdot \sigma_{std}(t_i) \right)^2 + \left( \frac{a}{a+b} \cdot \frac{6-i}{6} \cdot \sigma_{sub}(t_0) \right)^2$$

$$+ \left( \frac{a}{a+b} \cdot \frac{i}{6} \cdot \sigma_{sub}(t_6) \right)^2 + \left( \frac{b}{a+b} \cdot \sigma_{std}(t_j) \right)^2 + \left( \frac{b}{a+b} \cdot \frac{42-j}{6} \cdot \sigma_{sub}(t_{36}) \right)^2$$

$$+ \left( \frac{b}{a+b} \cdot \frac{j-36}{6} \cdot \sigma_{sub}(t_{42}) \right)^2 \tag{13}$$

where $a : b = (t_j - t_k) : (t_k - t_i)$. Hat "^" means estimated value. $\hat{V}_{sub}(t_k)$ was calculated by interpolating the output of the SWS-gas value nearest to the targeted sample as follows:

$\{spl1 \mid k = 4\}$

$$\hat{V}_{sub}(t_4) = \frac{1}{3} \cdot V_{sub}(t_0) + \frac{2}{3} \cdot V_{sub}(t_6) \tag{14}$$

$\{spl1 \mid k = 7, 10, 13, \cdots, 34\}$

$$\hat{V}_{sub}(t_k) = \frac{2}{3} \cdot V_{sub}(t_{k-1}) + \frac{1}{3} \cdot V_{sub}(t_{k+2}) \tag{15}$$

$\{spl2 \mid k = 5\}$

$$\hat{V}_{sub}(t_5) = \frac{1}{6} \cdot V_{sub}(t_0) + \frac{5}{6} \cdot V_{sub}(t_6) \tag{16}$$

$\{spl2 \mid k = 8, 11, 14, \cdots, 35\}$

$$\hat{V}_{sub}(t_k) = \frac{1}{3} \cdot V_{sub}(t_{k-2}) + \frac{2}{3} \cdot V_{sub}(t_{k+1}) \tag{17}$$

Here $V_{sub}(t_l)$ $\{sub \mid l = 0, 6, 9, 12, \cdots, 36\}$ is the measured value. In the following, a calculation example of $\hat{V}^{BE}_{std(t_i, t_j)}(t_k)$ and the variance for the case $\{spl1 \mid k = 4, (i, j)\}$ are given without any estimated value.

$\{spl1 \mid k = 4, (i, j) = (1,37), (2,38), (3,39)\}$

$$\hat{V}^{BE}_{std(t_i, t_j)}(t_4) = \frac{1}{3} \cdot V_{sub}(t_0) + \frac{2}{3} \cdot V_{sub}(t_6) + \frac{a}{a+b} \cdot \left\{ V_{std}(t_i) - \left( \frac{6-i}{6} \cdot V_{sub}(t_0) + \frac{i}{6} \cdot V_{sub}(t_6) \right) \right\}$$

$$+ \frac{b}{a+b} \cdot \left\{ V_{std}(t_j) - \left( \frac{42-j}{6} \cdot V_{sub}(t_{36}) + \frac{j-36}{6} \cdot V_{sub}(t_{42}) \right) \right\} \tag{18}$$

$$\left( \hat{\sigma}^{BE}_{std(t_i, t_j)}(t_4) \right)^2 = \left( \frac{1}{3} \cdot \sigma_{sub}(t_0) \right)^2 + \left( \frac{2}{3} \cdot \sigma_{sub}(t_6) \right)^2$$

$$+ \left( \frac{a}{a+b} \cdot \sigma_{std}(t_i) \right)^2 + \left( \frac{a}{a+b} \cdot \frac{6-i}{6} \cdot \sigma_{sub}(t_0) \right)^2 + \left( \frac{a}{a+b} \cdot \frac{i}{6} \cdot \sigma_{sub}(t_6) \right)^2$$

$$+ \left( \frac{b}{a+b} \cdot \sigma_{std}(t_j) \right)^2 + \left( \frac{b}{a+b} \cdot \frac{42-j}{6} \cdot \sigma_{sub}(t_{36}) \right)^2 + \left( \frac{b}{a+b} \cdot \frac{j-36}{6} \cdot \sigma_{sub}(t_{42}) \right)^2 \tag{19}$$


### 2.4.2 Estimation of sample air mole fraction and its combined standard uncertainty using a calibration line

We calculated a calibration line with the estimated outputs of standard gases ($\hat{V}^{BE}_{std(t_i, t_j)}(t_k)$) and their variances

($\left( \hat{\sigma}^{BE}_{std(t_i, t_j)}(t_k) \right)^2$) at the time of the sample measurement obtained in the section 2.4.1. Although the NDIR output may be

regressed with a polynomial equation (*e.g.,* Tanaka et al., 1983), there is no significant difference between the linear and
quadratic regression results in this system (shown in Section 2.5). To perform a weighted regression with the output of each
standard gas, a linear line ($y = Sx + I$; $y$: output in voltage, $x$: mole fraction for $CO_2$ and log(mole fraction) for $CH_4$) was
adopted for the calibration line (Fig. 4).

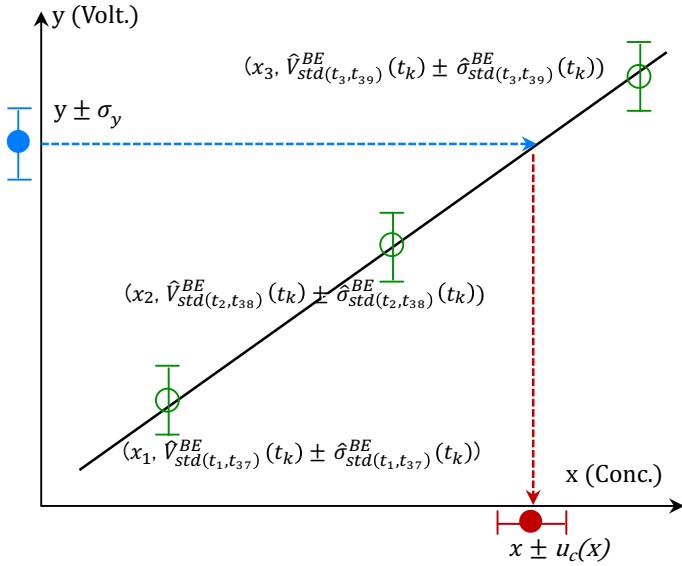

**Figure 4. Schematic diagram for estimating the $CO_2$ and $CH_4$ mole fraction ($x$) of the gases and the combined standard uncertainty**
**($u_c(x)$) from the output in voltage ($y$) with its SD ($\sigma_y$). The gray line indicates the estimated linear calibration line ($y = Sx + I$).**

Following the likelihood method, we identified the slope ($S$) and intercept ($I$) for every sample time ($k$) at the maximum of the

likelihood function ($L$). Solving the normal equation of $\begin{cases} \frac{\partial L}{\partial S} = 0 \\ \frac{\partial L}{\partial I} = 0 \end{cases}$, $S$ and $I$ were obtained as follows:

$$S(k) = \frac{(\sum w_{ijk})(\sum w_{ijk}x_i y_{ijk}) - (\sum w_{ijk}x_i)(\sum w_{ijk}y_{ijk})}{(\sum w_{ijk})(\sum w_{ijk}x_i^2) - (\sum w_{ijk}x_i)^2} \tag{20}$$

$$I(k) = \frac{(\sum w_{ijk}y_{ijk})(\sum w_{ijk}x_i^2) - (\sum w_{ijk}x_i y_{ijk})(\sum w_{ijk}x_i)}{(\sum w_{ijk})(\sum w_{ijk}x_i^2) - (\sum w_{ijk}x_i)^2} \tag{21}$$

where $x_i$ is WS-gas mole fraction determined against the NIES scale. $y_{ijk}$ is the estimated outputs of standard gas ($\hat{V}^{BE}_{std(t_i,t_j)}(t_k)$)

and $w_{ijk}$ is the reciprocal of the variance ($1/\left(\hat{\sigma}^{BE}_{std(t_i,t_j)}(t_k)\right)^2$). $\Sigma$ indicates the sum of $i$ (three standard gases) and the same

for the following discussion. As shown in Section 2.4.1, the combinations of (i, j) are (1, 37), (2, 38), and (3, 39). We omitted
$i, j,$ and $k$ for the following expression. The linearity of the calibration line was assessed using the correlation coefficient, and
data were rejected when the linearity was deemed insufficient.

The inverse function was used because we estimated the mole fraction from the output in voltage. Furthermore, because the calibration line passes through the weighted mean point $(\bar{x}, \bar{y}) = \left(\frac{\sum wx}{\sum w}, \frac{\sum wy}{\sum w}\right)$, we practically used the following line:

$$x = \frac{y - \bar{y}}{S} + \bar{x} \tag{22}$$

The square of combined standard uncertainty $(u_c(x))$ for the estimated mole fraction $(x)$ was calculated with the following equation (WMO, 2020):

$$u_c^2(x) = \left(\frac{\partial x}{\partial y}\right)^2 u^2(y) + \left(\frac{\partial x}{\partial \bar{y}}\right)^2 u^2(\bar{y}) + \left(\frac{\partial x}{\partial S}\right)^2 u^2(S) + \left(\frac{\partial x}{\partial \bar{x}}\right)^2 u^2(\bar{x}) \tag{23-1}$$

where $u^2$ is variance for each component. The first term expresses the contribution from the variation in output of the measured air $(\sigma_y)$ and 60 repeated measurements:

$$\left(\frac{\partial x}{\partial y}\right)^2 u^2(y) = \frac{\sigma_y^2}{S^2} \cdot \frac{1}{60}$$

The second term expresses the contribution from the variation in $\bar{y}$:

$$\left(\frac{\partial x}{\partial \bar{y}}\right)^2 u^2(\bar{y}) = \frac{1}{S^2} \frac{\sum w^2 \sigma^2}{(\sum w)^2} = \frac{1}{S^2} \cdot \frac{1}{\sum w}$$

where $\sigma^2$ is the variance of the output for standard gases constituting the calibration line. The third term expresses the contribution from the variation in the slope of the calibration line $(S)$:

$$\left(\frac{\partial x}{\partial S}\right)^2 u^2(S) = \frac{(y - \frac{\sum wy}{\sum w})^2}{S^4} \cdot \sum \sigma^2 \left(\frac{\partial S}{\partial y}\right)^2 = \frac{(y - \frac{\sum wy}{\sum w})^2}{S^4} \cdot \frac{\sum w}{(\sum w)(\sum wx^2) - (\sum wx)^2}$$

The fourth term expresses the contribution from the variation in $\bar{x}$. The NIES09 $CO_2$ scale is based on the gravimetric primary standard gases using a one-step dilution (Machida et al., 2011). When fifteen gravimetrically prepared mixed gases in the range of 350-390 ppm were measured on an existing scale, the SD of the differences from their mole fractions determined on that existing scale was 0.042 ppm (Tohjima et al., 2006), which we adopt as the uncertainty of the gravimetric method. For the NIES09 $CO_2$ scale, we prepared eight standard gases gravimetrically in the range of 340-450 ppm for atmospheric measurements. However, we maintain as primary standards of the NIES09 $CO_2$ scale the calibration results of eight different standard gases in the same concentration range that had demonstrated long-term stability, calibrated using the gravimetrically prepared standards (Machida et al., 2011). During calibration of primary standards, we performed repeated measurements (N=27-40) and used their mean values as calibration values. The SD of these measurements was 0.01-0.02 ppm (similar values were observed for calibrations of secondary and working standards). Since the mole fractions of standards prepared by one-step gravimetric method are independent, the uncertainty for each primary standard becomes 0.02 ppm ($\sqrt{\frac{0.042^2}{8} + SD^2/N}$), including the transfer uncertainty. In effect, as the propagation term is an order of magnitude smaller, the root of the fourth term in equation 23-1 also becomes 0.02 ppm.

On the one hand, the NIES94 CH$_4$ scale is based on the gravimetric primary standard gases using a four-step dilution. The root of the fourth term is estimated 3.2 ppb (Supplement). These values of the fourth term are common to all data points, so they should be denoted separately. Summarizing the first three terms, the $u_c(x)$ for the estimated mole fraction ($x$) is as follows:

$$u_c(x) = \frac{1}{S} \cdot \sqrt{\frac{\sigma_y^2}{60} + \frac{1}{\sum w} + \frac{1}{S^2} \cdot \frac{\left((\sum w)y - \sum wy\right)^2}{(\sum w)^2(\sum wx^2) - (\sum w)(\sum wx)^2}} \qquad (23\text{-}2)$$

Since the calculation was done for the logarithm of the mole fraction for CH$_4$, the $u_c(x)$ for the estimated mole fraction was determined differently for the higher level ($u^+ = x(e^{u_c(x)} - 1)$) and lower level ($u^- = x(1 - e^{-u_c(x)})$). However, the average value is expressed as the $u_c(x)$ since the difference is less than 0.1 ppb in real terms.

Figures S3 to S11 show the time series of the $u_c(x)$ for the ambient air CO$_2$ mole fraction. Most uncertainties were distributed around 0.05 ppm, but they can be higher than 0.3 ppm especially during summer. Note that the SD of the output ($\sigma_y$) of the sample air could become significant due to large diurnal variation during summer (Sasakawa et al., 2013) since the output of the sample air could include a natural variation of the atmosphere during the measurement period of three minutes. Therefore, following the recommendations outlined in the "Sensor precision and atmospheric variability" chapter of the WMO report (2020), it is valuable to concurrently present the sensor repeatability ($u_r$). The sensor repeatability should ideally be determined by conducting continuous measurements with a cylinder connected to the sample inlet. However, this approach has been impractical since the system's initial installation. Nevertheless, in our measurement system, SWS-gas measurements are conducted before and after each sample measurement, whereby constant gas from the cylinder is measured continuously for three minutes. We estimated $u_r$ by calculating the $u_c(x)$ for the SWS-gas following procedures analogous to those used for sample calculations and then temporally interpolating these values between the pre- and post-sample SWS-gas measurements over the sample measurement period. The estimated $u_r$ were distributed so as to closely follow the minimum values of sample $u_c(x)$ (Figs. S3-S11). In rare cases, they were slightly higher than the sample minimum values, which occurred when the mole fraction of the SWS-gas exceeded that of the highest WS-gas, resulting in increased uncertainty from the calibration curve.

Figures S12 to S17 show the time series of the $u_c(x)$ for the ambient air CH$_4$ mole fraction. Most are within five ppb, but they can be above ten ppb during summer at KRS and DEM. Meanwhile, at NOY, although the baseline is below five ppb, values exceeding ten ppb can occur regardless of season. All sites are affected by short-term variations from summer wetland emissions, while NOY is presumed to experience additional significant short-term variations from anthropogenic methane sources such as leakage of natural gas. The $u_r$ was estimated using a method similar to that for CO$_2$, and the values were distributed so as to closely follow the minimum values of sample $u_c(x)$. Although values remained below five ppb, they could fluctuate from near zero to five ppb over several days. Such fluctuations were also apparent in the samples and are considered to be mainly due to sensor stability.

## 2.5 Stability check with the SWS-gas measurement

We calculated a calibration line only when the SWS-gas measurements closest to both sides of the sample measurements were normal. The normality of the SWS-gas measurement was assessed as follows. The same on-site compressed air was measured

several times (for about a week) since the air was used as SWS-gas until its pressure dropped below 0.1 MPa. The on-site compressed air output value would vary smoothly if the system were stable. When the system temporarily became unstable the corresponding large changes of the analyzer output could not be corrected by the SWS to a sufficient degree. To identify such occasions, we first estimated the output of standard gases at the time of the target SWS-gas measurement by interpolating

$\Delta^B(t_i)$ and $\Delta^E(t_j)$ based on the output value of the target SWS-gas itself (Section 2.4.1). Then, calculating a calibration line with the estimated output of standard gases, we obtained the mole fraction of the target SWS-gas. Second, we estimated the output value of the SWS-gas at the time of the target SWS-gas measurement by interpolating the outputs of the two adjacent SWS-gases. Then, we determined the mole fraction of the target SWS-gas in the same manner. If these estimated mole fractions differed from each other by more than one ppm for $CO_2$ and ten ppb for $CH_4$, we regarded the target SWS-gas data as abnormal.

This assessment (referred to as "self-check-value (scv) for SWS-gases" in Fig. 5) was done while the adjacent SWS-gas measurements were conducted for the same on-site compressed air.

We then checked the system's stability with the measurement of the SWS-gas. Interpolating the outputs of the SWS-gases adjacent to the standard gas measurements, we calculated the mole fractions of the SWS-gas with the calibration line at the time of STD2, which was used to assess the coefficient of determination in Section 2.3. We regarded the estimated mole

fractions of the SWS-gas as independent; thus, we obtained 14 estimated mole fractions if the measurements for the same SWS-gas continued for a week. We determined a threshold for the SD ($\sigma_{sws}$; 1 ppm for $CO_2$, 10 ppb for $CH_4$) and the fluctuation range (3 ppm for $CO_2$ and 30 ppb for $CH_4$) obtained from the estimated independent data set. All the data that exceeded the threshold were deleted. A flow chart for the calculation method is shown in Fig. 5.

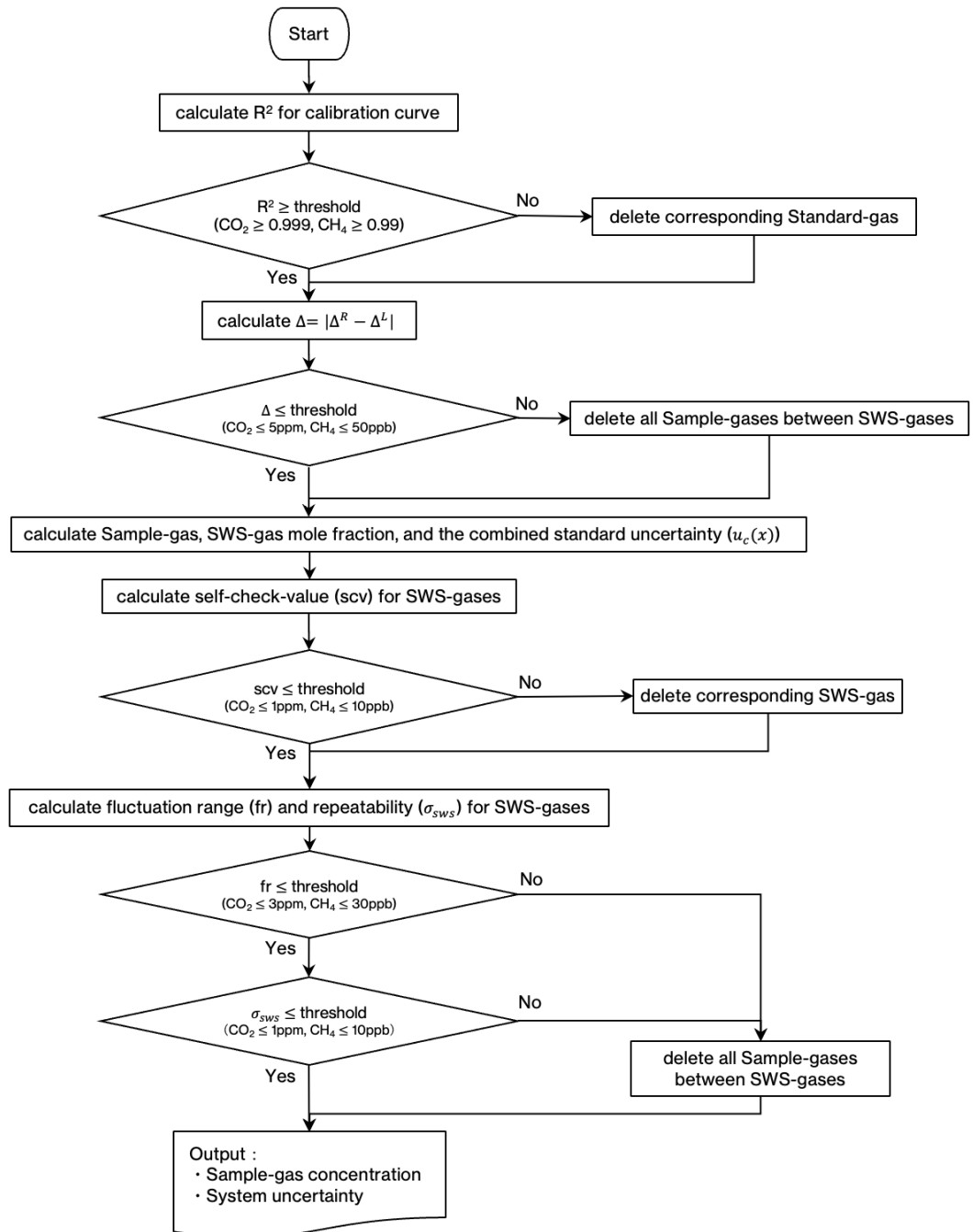

**Figure 5. Flow chart of mole fraction calculation method.**

## 3 Reproducibility

### 3.1 Change in temporal conditions

The $\sigma_{sws}$ that remained after meeting the criteria described in Section 2.5 serve as an indicator of measurement reproducibility
at 12-hour intervals throughout the given period (often a week). Regardless of the site or time of measurement, $\sigma_{sws}$ for $CO_2$ mostly was below 0.2 ppm, and it was below five ppb for $CH_4$ (Supplement Figs. S3-S17). After introducing the CRDS, the working standard gas measurement interval was changed to 48 hours. At the same time, since the consumption flow rate has doubled, the SWS-gas changeover time has decreased to approximately three days, which makes the number of SWS measurements with NDIR/TOS only three times at most. Although the $\sigma_{sws}$ values obtained from a few data are for reference
only, they were distributed in almost the same range (light blue dots in Figs. S3-S17).

### 3.2 Change in sensor and temporal conditions

The CRDS operated at KRS from July 2015, DEM from June 2016, and NOY from August 2016, albeit for a short period of time for JR-STATION. The CRDS is a highly stable analyzer used in greenhouse gas observations worldwide (Kwok et al., 2015). We compared the recalculated NDIR (and TOS) values with the CRDS values to check the long-term reproducibility
(Figs. 6-7, Supplement Figs. S18-S21). The flow path branches off after the 6-port valve (Fig. 1), so the same air is analyzed. The CRDS operates independently of the existing system, so information on instrument error flags and valve switching timing is not linked to the measured data. Therefore, the timing of the standard gas measurement was captured by detecting the $CH_4$ mole fraction of the lowest standard gas. Only data from periods was extracted when the SWS results fulfilled the criteria outlined in the previous section. The CRDS output values were converted to the NIES scale based on the WS measurements
and averaged over three minutes for comparison. The temperature in the warm box of CRDS (data column name is 'WarmBoxTemperature') was kept constant (45.00°C), but it may vary significantly in rare cases. The CRDS outputs data every few seconds, but the output frequency may drop abnormally. Since the device is considered unstable under these conditions, the data was not used for comparison if the temperature change was more than 0.03ºC or if the number of data was less than ten in three minutes averaged over the CRDS data. Since some observed values fell outside the mole fraction range
of the standard gas, only values within the range were used to calculate the difference between the two.

As for the NDIR, there was no significant difference between the high-inlet and low-inlet differences, indicating no bias due to differences in inlets (Fig. 6, Supplement Figs. S18-S20). However, regardless of the year of observation, the NDIR showed lower values than those of the CRDS by about 0.1 ppm at all sites. $CO_2$ mole fraction has more significant diurnal variation during the summer months (Sasakawa et al., 2013), so the error bars were more prominent (Supplement Fig. S20). Still, the
amount of bias remains the same. As mentioned in Section 2.1, using our isotopically lighter standard gas, the NDIR measures a lower $CO_2$ mole fraction in a sample air with an actual $CO_2$ mole fraction. This 0.1 ppm difference can be attributed to the optical filter property of the NDIR used in this system. In addition, this bias does not change over time, indicating that this

system produces results with good reproducibility over time. We plan to make the correction of 0.1 ppm for the NDIR in the following published data.

Results from data using calibration curves with quadratic equations are also shown. There were no significant differences from the results using the linear regression in any concentration range, confirming the linearity of the NDIR used in this system. In the first place, the coefficients of the second order of the regressed quadratic equation frequently changed, both positively and negatively, so it is unlikely that the essential response of NIDR can be estimated by a polynomial equation in the concentration range of the standard gas used in this system.

As for the TOS, there was also no significant difference between the high-inlet and low-inlet differences, indicating no bias due to differences in inlets (Fig. 7). As an overall average within the actual measurement range in Siberia, the TOS did not differ from the CRDS, but there can be a bias from $-10\pm5$ ppb to $5\pm5$ ppb with the CRDS output, depending on the mole fraction. However, the degree of the bias varied from year to year, and its cause was unknown, so a constant correction cannot be made. The time series showed higher values during the winter period in some years (Supplement Fig. S21), which may be

the result of the increase in the mole fraction-dependent difference seen in Fig. 7.

At NOY and DEM, it was discovered that the temperature controller of the catalytic unit was not functioning correctly. Since the TOS is sensitive to CO and $H_2$ in the air, it could produce unusually high values without a proper catalytic unit. For the period, only the data from the CRDS should be published. Since no catalytic unit errors were identified at the other sites, the ambient atmospheric values were detected, as is the case with KRS.

Both the NDIR and the TOS have shown good reproducibility over several years for the CRDS, suggesting that this measurement is convincing.

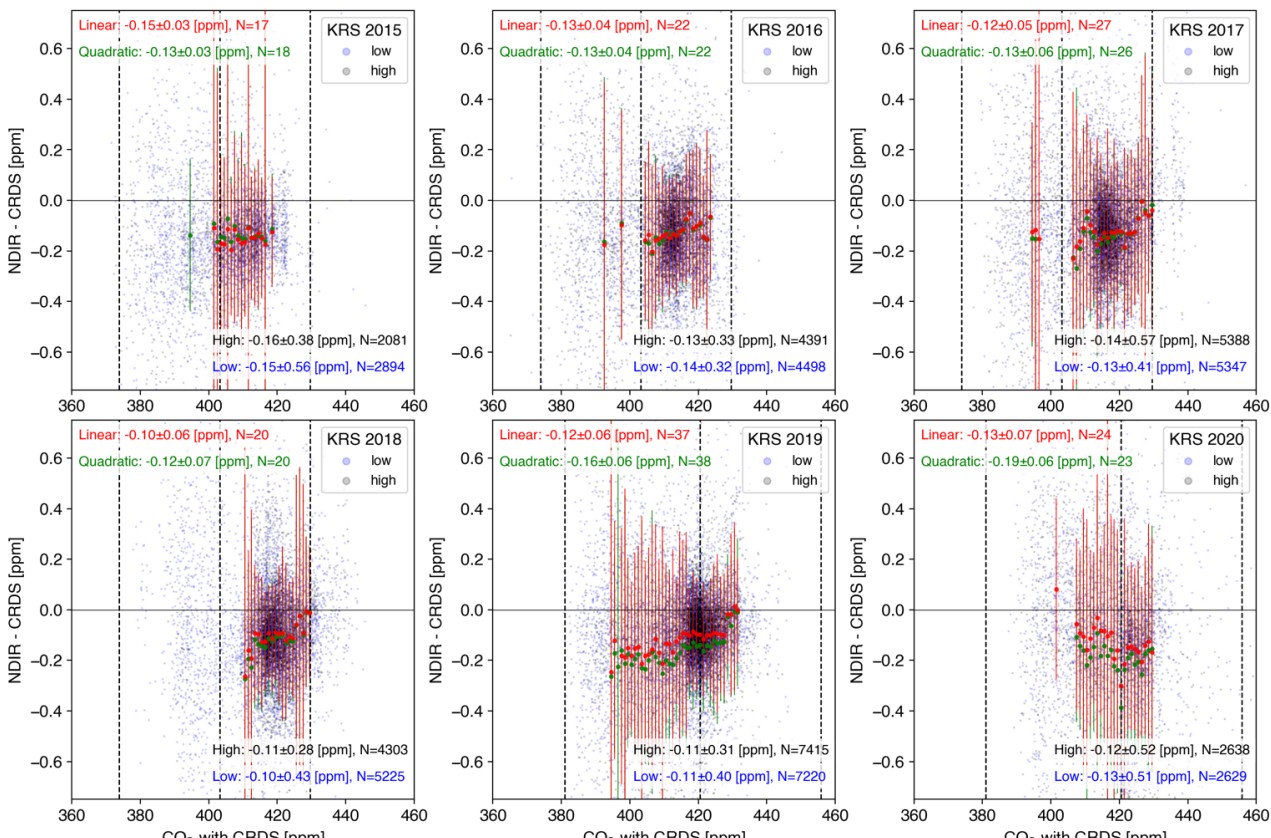

**Figure 6. Relationship between the CO₂ mole fraction by the CRDS at KRS and the difference in respective CO₂ mole fractions measured by the NDIR and the CRDS (NDIR – CRDS): The CRDS values were averaged over the corresponding 3-minute period. The light blue (gray) circle shows the difference from a low (high) altitude inlet. The dotted lines indicate the mole fraction of standard gases. The figure (mean ± SD) in the right bottom represents the average difference for each inlet. Only data that were within the standard gas mole fraction range were used. The red dots indicate the values averaged every one ppm for the combined high-altitude and low-altitude data. However, calculations were only made when the number of data points used was 100 or more. The results from the calibration curve using the quadratic equation are shown as green dots. Error bars indicate the SD. These annual averages are marked by their respective colors in the upper left.**

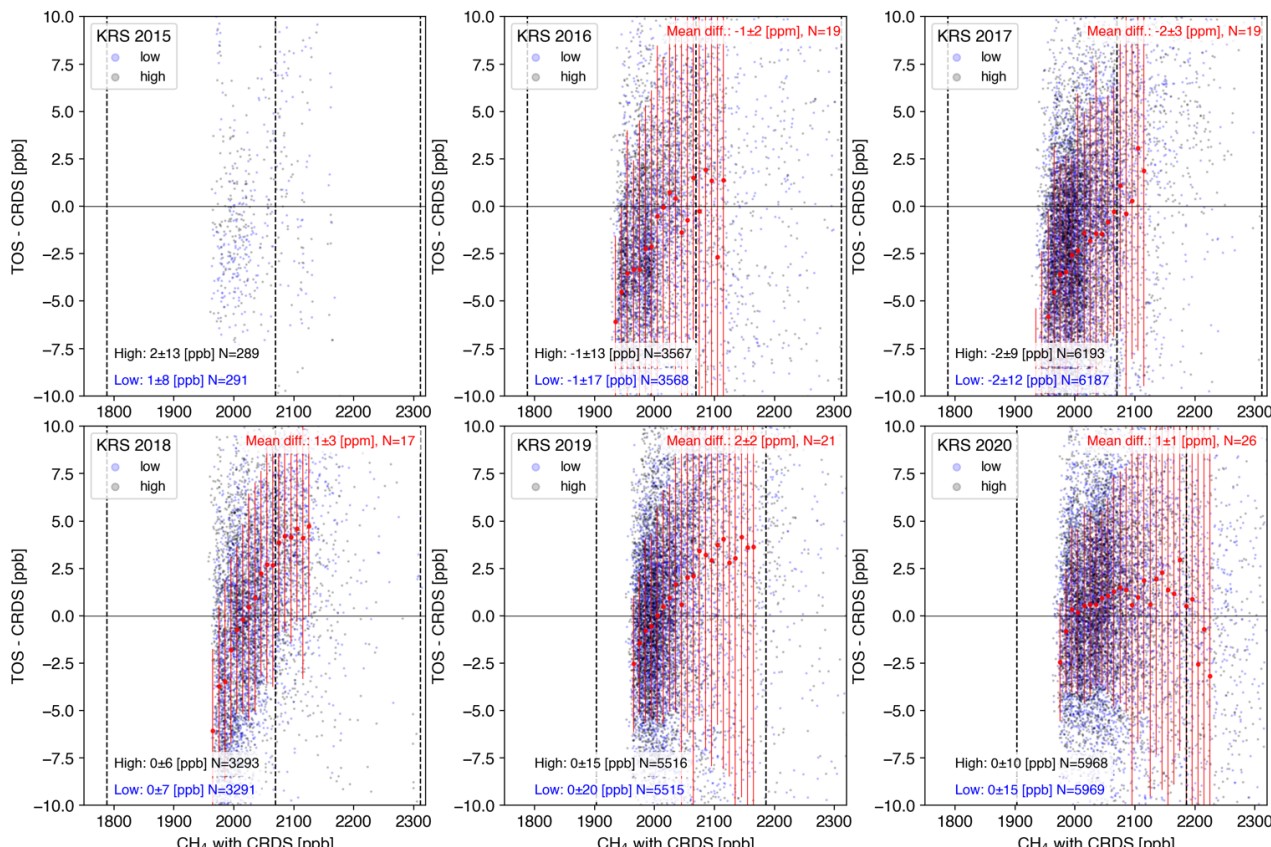

**Figure 7. Relationship between the CH₄ mole fraction by the CRDS at KRS and the difference in respective CH₄ mole fractions measured by the TOS and the CRDS (TOS – CRDS): The CRDS values were averaged over the corresponding 3-minute period. The light blue (gray) circle shows the difference from a low (high) altitude inlet. The dotted lines indicate the mole fraction of standard gases. The figure (mean ± SD) in the left bottom represents the average difference for each inlet. Only data that were within the standard gas mole fraction range were used. The red dots indicate the values averaged every ten ppb for the combined high-altitude and low-altitude data. However, calculations were only made when the number of data points used was 100 or more. Error bars indicate the SD. The annual average is marked in the upper right.**

## 4 Full uncertainty estimates

In Table 3, we present all uncertainties estimated in this study. Since our measurements are based on gravimetrically prepared primary standards, we have also calculated the uncertainties propagated from the preparation of primary standards through to working standard gases (see Section 2.4.2 and Supplement). Our observation system records three-minute average mole fractions of atmospheric $CO_2$ and $CH_4$ at multiple heights every hour. These average values are published with their $u_c(x)$. While our system corrects for sensor drift using on-site air (SWS-gas), we estimated the sensor repeatability ($u_r$) during three-

minute sample measurements using the $u_c(x)$ of three-minute measurements of this SWS-gas (Section 2.4.2). The SWS-gas was used for approximately one week, and its reproducibility was estimated from the repeated measurements (Section 3.1). We verified reproducibility for three sites where the CRDS was installed by comparing the CRDS and the NDIR (or TOS)

measurements (Section 3.2). For $CO_2$, the NDIR showed 0.1 ppm lower values, which we attributed to isotope effects in NDIR; consequently, we plan to implement corrections for NDIR measurements. For $CH_4$, annual mean values agreed within the measurement scatter. However, within the range of mole fractions measured each year, differences from $-10\pm5$ ppb to $5\pm5$ ppb occurred.

*Table 3.* Summary of uncertainties.

| | $CO_2$ (ppm) | $CH_4$ (ppb) | Method |
|---|---|---|---|
| Scale uncertainty | 0.02 | 3.2 | See Section 2.4.2 and Supplement. |
| Sensor repeatability | 0.05 | 5 | $u_c(x)$ for the SWS-gas meas. in 3 min. |
| Reproducibility (Time) | <0.2 | <5 | SD of SWS-gas measurements for one week. |
| Reproducibility (Sensor, Time) | $0.1^1$ | $-2\pm3$ (min.) to $2\pm2$ (max.) as annual mean<br><br>$-10\pm5$ (min.) to $5\pm5$ (max.) as interval mean at every 10 ppb | Difference in NDIR (or TOS) values from CRDS values (Figs.6-7). |

[1] Isotope effect in the NDIR. We plan to apply this correction to the NDIR measurements.

**5 Conclusions**

Using the standard gas saving system, we have developed and validated an improved methodology for calculating mole

fractions and uncertainties in long-term $CO_2$ and $CH_4$ measurements at remote sites. Following the recommendations shown in the GAW report (WMO, 2020), we provided both the combined standard uncertainty ($u_c(x)$) and sensor repeatability ($u_r$) for each measurement, enabling data users to select appropriate data based on their specific research requirements. Comparison with CRDS measurements demonstrated excellent long-term reproducibility of the NDIR and TOS, with the NDIR showing a consistent offset due to isotope effects that can be corrected. For $CH_4$, while annual means agreed well with CRDS, some

differences were observed depending on the year and mole fraction level. The working standard gas saving system, using on-site compressed air as sub-working standard gas, proved effective for long-term monitoring at remote sites, showing good temporal reproducibility. By quantifying all major uncertainty components, including those from primary standard gas preparation to working standards, we have provided a comprehensive uncertainty budget that enhances the value of the JR-STATION dataset for studying greenhouse gas dynamics in Siberia. The validated performance of our cost-effective system

design demonstrates its suitability for maintaining high-quality, long-term greenhouse gas monitoring networks in remote regions where infrastructure is limited.

## Data availability

The data are available from the Global Environmental Database, hosted by ESD, NIES (http://db.cger.nies.go.jp/portal/geds/index).

## Author contribution

MS and NT designed the study. MS wrote the manuscript. MA, DD, and AF conducted the measurements. All authors contributed to the discussion and preparation of the manuscript.

## Competing interests

The authors declare that they have no conflict of interest.

## Acknowledgments

We thank Sergey Mitin (Mikkom Ltd.) for his long-term administrative support since before the initiation of these observations. We are grateful to the two reviewers whose insightful comments significantly improved this manuscript. Our research was financially supported by the Global Environmental Research Coordination System from Ministry of the Environment of Japan (E0752, E1254, E1752, E2251) and the most important innovative project of national importance "Development of a system for ground-based and remote monitoring of carbon pools and greenhouse gas fluxes in the territory of the Russian Federation, ensuring the creation of recording data systems on the fluxes of climate-active substances and the carbon budget in forests and other terrestrial ecological systems" (Registration number: 123030300031-6).

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
