# Peer review of "Revised methodology for CO2 and CH4 measurements at remote sites using a working standard gas saving system"

_Atmospheric Measurement Techniques, 2023_

## Author Response (AR3)

Reply to the reviewer's comments.

Upon reviewing the GAW report (2020), we comprehensively revised our uncertainty-related terminology. Specifically, we noted that the equation on page 20 of the report is identical to Equation 23-2 in our manuscript, thus we now refer to this as "combined standard uncertainty." Furthermore, page 20 of the report provides the following definition of "reproducibility":

"Reproducibility is a measure of the closeness of agreement of measurements carried out under changed conditions of the measurement. Changed conditions may include new sensors, new measurement principle, new reference standard(s), new location, and/or time. It is strongly recommended to regularly analyse a target gas to estimate reproducibility, even if this measurement does not cover all sources of uncertainty"

Consequently, we revised our terminology as follows: The SD derived from SWS-gas measurements described in Section 2.4.3 (new Section 3.1) represents reproducibility under changed temporal conditions and has been labeled accordingly. The results from the comparison with CRDS in Section 2.5 (new Section 3.2) represent reproducibility under changed sensor and temporal conditions and have been modified to reflect this.

Comment;

1. Authors reply:

We have added the following description in Section 2.4.2.

"The NIES09 scale is based on the gravimetric primary standard gases using a one-step dilution (Tohjima et al., 2006), and its overall uncertainty, including transfer, is estimated to be 0.043 ppm (Machida et al., 2011). This value is equal to , but this term is not included in the calculation because cannot be determined precisely; as for the value of , it distributed between 0.025 ppm to 0.043 ppm, since there are three WSs."

a) This only addresses CO2. It does not give information on the CH4 propagation uncertainty. b) In Tohjima et al 2006 on the one hand side an uncertainty estimate is made for the methodology that the NIES 09 scale is based on that results in an uncertainty of 0.032 ppm (based on uncertainties from gas purities, mass measurements etc.). The associated uncertainty terms would likely bias all primary standards prepared by the method in the same way and would not introduce scatter within a set of standards prepared in the same way. This very scatter of the deviations for individual primary standards relative to a common reference (NIES95 scale) is also provided in Tohjima et al. 2006 as 0.042 ppm for 15 individual standards produced in three batches each a few months apart. This number then has been

quoted as "preparing uncertainty of one-step dilution cylinder" by Machida et al, 2011. Machida et al. 2011 quote an additional uncertainty for the scale transfer to the set of scale secondaries with the "uncertainty in standard transfer process by our NDIR calibration system was estimated to be 0.010 ppm". This scale propagation was essential for maintaining the NIES 09 scale (CO2 in the primary standard cylinders was suspected to drift) so it is clear that utmost attention has been spent for this transfer. I wonder if the same effort has been spent each time in assigning the Siberian WS. Machida et al, 2011 also explain that comparison between the scale secondaries and another set resulted in variable results that were within  $\pm$  0.06 ppm. Therefore, perhaps this might be a more realistic uncertainty estimate to quote for the assignment of the WS (in addition to the scale uncertainty)? The uncertainty that the WMO-CCL has estimated for assignments of standards using the NDIR technology is 0.03 ppm that they came up with based on very thorough quality control documentation. Achieving a measurement uncertainty of 0.01 ppm for assignment of standards over a long period using NDIR appears like an extremely challenging task. Therefore, I feel that such an estimate requires to present more evidence than the reference of the 2009 meeting report.

The following sentences (starting l. 265) are confusing to me "The production of standard gases is typically conducted concurrently using the same equipment (such as dilution vessels and pressure gauges). This process may introduce common biases, albeit to varying degrees, potentially resulting in correlated (non-independent) mole fractions." My thinking has been that the JP-Siberian WS were assigned by NDIR comparison measurements vs. the NIES09 scale standards and I do not understand what the referred hardware (as dilution vessels and pressure gauges) should add as bias.

In my understanding the scale uncertainty of 0.043 ppm is specifying the uncertainty estimate made for the scale and thus cannot be reduced by using multiple standards at stations. As stated before I assume there needs to be an additional scale propagation uncertainty term be accounted for each standard assignment and the number of calibration standards employed then will reduce the uncertainty.

**Reply;**

The value of  $\pm 0.06$  ppm shown in Machida et al. (2011) represents a drift amount over one and a half years and is not an appropriate value to express uncertainty. As pointed out, scale propagation uncertainty needs to be added to the WS uncertainty. Based on the details of primary gas preparation, we have revised as follows:

"The fourth term expresses the contribution from the variation in  $\bar{x}$ . The NIES09 CO2 scale is based on the gravimetric primary standard gases using a one-step dilution (Machida et al., 2011). When fifteen gravimetrically prepared mixed gases in the range of 350-390 ppm were measured on an existing scale, the SD of the differences from their mole fractions determined on that existing scale was 0.042 ppm (Tohjima et al., 2006), which we adopt as the uncertainty of the gravimetric method. For the NIES09 CO2 scale, we prepared eight standard gases gravimetrically in the range of 340-450 ppm for atmospheric measurements. However, we maintain as primary standards of the NIES09 CO2 scale the calibration results of eight different standard gases in the same concentration range that had demonstrated long-term stability, calibrated using the gravimetrically prepared standards (Machida et al., 2011). During calibration of primary standards, we performed repeated measurements (N=27-40) and used their mean values as calibration values. The SD of these measurements was 0.01-0.02 ppm (similar values were observed for calibrations of secondary and working standards). Since the mole fractions of standards prepared by one-step gravimetric method are independent, the uncertainty for each primary standard becomes 0.02 ppm ( $\sqrt{\frac{0.042}{8} + SD^2/N}$ ), including the transfer uncertainty. In effect, as the propagation term is an order of magnitude smaller, the root of the fourth term in equation 23-1 also becomes 0.02 ppm."

The previous statement is misleading and has been removed. Regarding CH4, the root of the fourth term is estimated 3.2 ppb. Details have been added to the supplementary information.

**Comment;**

**2. Authors reply;**

The NIES CO2 scale agreed with the WMO scale to within 0.1 ppm, even with the results of Round Robin 5, which was measured in 2009 at NIES. This point was added in addition to CH4 scale. The results for Round Robin 4 are not shown as they were with a previous scale (NIES 95 scale). No other published data are available to directly compare the scales.

Machida et al. 2011 provide a NIES 95 – NIES09 scale conversion function for CO2 with the following statement "The uncertainty of CO2 calculation using equation (2) is 0.01 ppm". CO2[09] (ppm)= 1.0519x10-4x(CO2[95])2 + 0.922433xCO2[95] + 14.313So it is not clear why NIES95 data are assumed to be not available for a comparison.

**Reply;**

Thanks for your comment. The NIES 95 scale can indeed be converted to the 09 scale. When converted, 355 ppm and 385 ppm results still agree within 0.1 ppm, while 368 ppm results in

a difference of 0.14 ppm. However, in RR4, the WMO scale was also the previous X2005 scale. Furthermore, since the range of concentrations compared in RR4 is lower than those observed in the tower network, we do not believe it is necessary to present that result in the text.

**Comment;**

**3. Authors reply:**

NDIR calibrations were performed by linear regression, as shown in Figure 5, where the coefficient of determination is greater than or equal to 0.999; if it is less than this, the data is rejected. We have replaced the correlation plots with residual plots (Figure 6), which show linearity on the NDIR, taking the CRDS as a linear reference.

While the rejection criterion of 0.999 ensures that bad calibrations are filtered out, it does not mean that there is no remaining uncertainty from a linear regression fit that is caused by a non-linearity of the response. The Figure 6 and the corresponding figures S18 and S19 in the supplement material are useful. Yet, it is not possible to confirm or invalidate a systematic mole fraction dependent bias resulting from the assumption of a linear NDIR response based on the single data 3-minute average comparison plots. Those plots are too busy and the y-axis scale too large. That might perhaps become visible if a quadratic fit was made through the data points presented in Fig. 6, S18, S19 and could be quantified from that. Else an estimate for the uncertainty caused by application of a linear response curve based on laboratory or literature results would also be appropriate. Machida et al 2011 applied a cubic fit curve to a multipoint NDIR calibration.

As the authors state most of the scatter of the difference between the two analysers measurement data will be due to the atmospheric variability. That should cancel out, though, and there is a systematic -0.1 to -0.15 ppm CO2 discrepancy (NDIR-CRDS) at all sites and all years that are displayed. I do not follow the authors conclusion: "There was no significant difference in CO2 mole fraction regardless of the inlet at both altitudes (Fig. 6, Supplement Figs. S18-S20)." just because the mean absolute offset is less than the SD in those figures because the SD will mostly be resulting from the atmospheric variability and several thousand data points this will be significant.

**Reply;**

Thanks for the comment. I have modified Fig. 6, S18, and S19 also to show the results of the quadratic equation. The difference is not significant. The coefficients of the second order of

the quadratic equation frequently changed to positive and negative, so it is unlikely that this is a property of NDIR, at least within the range of the standard gases used (the percentage of positive coefficients varied from 20% to 70% from year to year). Machida et al. (2011) made a calibration curve over a wide concentration range of 110 ppm (340~450 ppm). On the other hand, the concentration range of this system is 60~70 ppm, so we believe it was possible to make a linear approximation. In addition, a more reliable calibration curve is created by weighting the inverse of the variance of each standard gas. Still, since there are three standard gases, the calibration curve is uniquely determined by a quadratic equation, and this calculation curve. The following sentences were added in the text before "A linear line..." in the first paragraph of Section 2.4.2 and in Section 2.5 (new Section 3.2).

"Although the NDIR output may be regressed with a polynomial equation (*e.g.,* Tanaka et al., 1983), there is no significant difference between the linear and quadratic regression results in this system (shown in Section 2.5). To perform a weighted regression with the output of each standard gas,"

"Results from data using calibration curves with quadratic equations are also shown. There were no significant differences from the results using the linear regression in any concentration range, confirming the linearity of the NDIR used in this system. In the first place, the coefficients of the second order of the regressed quadratic equation frequently changed, both positively and negatively, so it is unlikely that the essential response of NIDR can be estimated by a polynomial equation in the concentration range of the standard gas used in this system."

Indeed, the difference between the NDIR and the CRDS measurements was meaningful. As shown in the previously revised Section 2.1, when the NDIR analyzer is calibrated against  $CO_2$  standards with lighter-than-atmospheric  $CO_2$  isotopic composition, the NDIR analyzer measures a lower  $CO_2$  mole fraction in a sample air with a known  $CO_2$  mole fraction. However, the apparent NDIR  $CO_2$  mole fraction error depends on its individual sensitivity to the optical filter property. As can be seen from the revised Fig. 6, S18, and S19, the NDIR used in this system is expected to show a characteristic of about -0.1 ppm, so we decided to make this correction in the next published data. The text has been corrected as follows.

"There was no significant difference in  $CO_2$  mole fraction regardless of the inlet at both altitudes (Fig. 6, Supplement Figs. S18-S20)."

"As for the NDIR, there was no significant difference between the high-inlet and low-inlet differences, indicating no bias due to differences in inlets (Fig. 6, Supplement Figs. S18-S20). However, regardless of the year of observation, the NDIR showed lower values than those of the CRDS by about 0.1 ppm at all sites.  $CO_2$  mole fraction has more significant diurnal variation during the summer months (Sawakawa et al., 2013), so the error bars were more prominent (Supplement Fig. S20). Still, the amount of bias remains the same. As mentioned in Section 2.1, using our isotopically lighter standard gas, the NDIR measures a lower  $CO_2$  mole fraction in a sample air with an actual  $CO_2$  mole fraction. This 0.1 ppm difference can be attributed to the optical filter property of the NDIR used in this system. In addition, this bias does not change over time, indicating that this system produces results with good reproducibility over time. We plan to make the correction of 0.1 ppm for the NDIR in the following published data."

**Comment;**

**4. Authors reply:**

We have added the figures for differences vs time in the Supplement (Figures S20 and S21), which show a good agreement.

These figures are a good way to provide that information. To display the longer term reproducibility it would be helpful to aggregate them (e.g. have daily means of the interinstrument differences and the std.dev. of those daily data points) and have a smaller y-axis range instead (e.g. by factor of ten). The large y-axis scale range currently limits the information content considerably.

**Reply;**

At the first review round, the Reviewer #1 gave us the comment:

"A stronger independent check is the parallel measurement by the CRDS instruments. It is very convincing that the two measurements agree within the expectation and the annual mean differences remain pretty constant. I propose to add another figure for the Supplement displaying the NDIR-CRDS differences and TOS-CRDS differences vs time."

Although we showed the time series according to this comment, the values shown in Figs 6, 7, S18, and S19, which were re-created simultaneously, indicate that the annual mean differences were constant, so even without Figs. S20 and S21, the results were already convincing.

->

Fig. S20, with the modifications proposed here, is essentially the same as those shown in Fig. 6, except that  $CO_2$  has greater diurnal variation in the summer months, so the error bars are larger, but the amount of bias (due to isotope effects in the NDIR as described above) remains the same.

As for  $CH_4$ , in some years, the difference seems to be larger in winter (Fig. S21), but this is also explained already in Fig. 7, which shows the difference is mole-fraction-dependent. In any case, the reproducibility is good over a long period. This comparison shows that this measurement is convincing. Significant additions were made to Chapter 2.5 (new Chapter 3.2).

**Comment;**

**5. Authors reply:**

We have added the following sentences in Section 2.4.3.

"On the other hand, the CO2 (CH4) concentrations in these observations can fluctuate on the order of ppm (10 ppb), even during a few hours of daytime when the atmosphere is well mixed (Sasakawa et al., 2010, 2013). It is, therefore, considered adequate for observations carried out in the vicinity of strong emission and absorption sources, such as those in the Siberian interior. The GAW report states that the compatibility goal for CO2 (CH4) in the Northern Hemisphere is 0.1 ppm (2 ppb), but this is a target for background sites such as coastal areas and does not need to be strictly adapted to an observation area such as this study."

The GAW compatibility goals shall incorporate all uncertainties of the atmospheric measurements (with atmospheric variability not being part of this) and that is something that the manuscript has not yet fully provided. So I still see the usefulness of a summary of all uncertainties that combines assignment uncertainties, method repeatability (as approximated by sigma SWS), a non-linearity uncertainty estimate, isotopologue sensitivity uncertainty, …

**Reply;**

The text quoted in the Authors' reply was from our initial response and had already been largely revised in our previous modification. Additionally, the GAW report (2020) states on page 4 that "Network compatibility goal can only be assessed by comparing measurements of ambient air at a common site..." Upon recognizing that it would be inappropriate to derive network compatibility from the repeatability measurements discussed here, we have removed the related content. Regarding the recalculation of uncertainties, please refer to our subsequent responses.

Following the reviewer's comments, we added a new section titled "Summary of uncertainties" at the end. We also created a new section on "Reproducibility" and revised the overall structure.

**Comment;**

**6. Authors reply:**

"As for the isotope effect, Tohjima et al. (2009) reported that the errors for isotopic influence for the three NDIR analyzers range from -0.04 to -0.08 ppm. However, the apparent NDIR CO2 mole fraction error depends on its individual sensitivity to the optical filter property. The installation of this observation system started in the early 2000s, but the characteristics of each NDIR are not known, as this was not yet taken into account at that time."

In order to check each optical filter property, we need to get them back to our laboratory in Japan, which is impossible. So, we have described the possible bias: "The CO2 values measured by the system may appear low in the range of up to 0.08 ppm."

While more depleted reference standards compared to the atmospheric samples will cause a bias, the referenced publication (Tohjima et al., 2009) also shows that different individual analysers have slightly different isotopologue discriminations. So there will also be this different behaviour between the calibration NDIR instrument in the NIES lab and the analyzer in the field. The apparent differences between three such NDIR analyzers presented in that 2009 publication suggests an additional 0.07 ppm uncertainty resulting from this.

**Reply;**

In the previous revision, the most considerable value of 0.08 ppm among those shown by Tohjima et al. (2009) was described as a possible bias because we knew that the NDIR optical filters have individual characteristics. However, in the course of our analysis based on the comments, we found that the isotope effect was about 0.1 ppm based on a comparison with the CRDS, so we have revised the text as described above and will correct this in the next release of the data. All WS-gas calibrations performed in the laboratory at NIES were performed with depleted isotope composition standard gases, so the isotope effect in the process can be neglected.

**Comment;**

**7. Authors reply:**

The uncertainty of the analysis system is expressed in sigma  $\sigma$ sws using SWS-gas, as denoted

in Section 2.4.3, not here. As required by each element of equation (22), equation 1.249 is part of the uncertainty in the estimated concentration (x), including atmospheric variability over a 3-minute period. We consider this equation to be appropriate.

I guess I do not fully understand this but the previous suggestion might also have been misunderstood. My point was to rather not use the 3-min-atmospheric-variability for the first term of equation (23-1) but rather approximate the detector's 3 min measurement scatter by the respective 3-min-SD of the weighted SWS SD in order to not include the atmospheric variability (following the advice of GAW report 255).

**Reply;**

We acknowledge our misinterpretation. Following your suggestion, we have re-estimated the uncertainty, excluding atmospheric variability, using the values from pre- and post-sample SWS measurements. This analysis has been added to Section 2.4.2 following Equation 23-2.

**Comment;**

8. Authors reply:

The WBT-based data selection has been indicated in Section 2.5. It is now not shown in the Figures.

Please introduce the acronym WBT (WarmBoxTemperature) somewhere in the manuscript. This is missing.

**Reply;**

The current version should not use the term "WBT". If it is still there, please point it out so that we can remove it.

**Comment;**

One small detail:

All Figures S4-S17 refer in their caption to "The same Figure as Figure S1…" this should rather be S3.

**Reply; We have corrected them.**

---

## Author Response (AR4)

Reply to the editor's comments.

Thank you for coordinating the review process and for your comments. Our responses to your comments are as follows.

Comment;

In the abstract "The combined standard uncertainties $(u_c(x))$ of time-averaged data.." should be improved with "The combined standard uncertainties $(u_c(x))$ of time-averaged ambient air measurements.." - please check/agree.

Reply;

We have corrected it.

Comment;

Your new sections on Reproducibility and Summary of uncertainties:

Suggest "Full uncertainty estimates" rather than "Summary of uncertainties". 3.1 and 3.2 condition or conditions? Probably "conditions".

Reply;

We have replaced them with "Full uncertainty estimates" and "conditions".

Comment;

The new Table 3: What does "~" mean? Please use more words to make your calculations understandable. e.g. NDIR (SnO2) – CRDS: Does this mean the calculated difference between the measurements by these systems?

Reply;

The following corrections have been made.

*Table 3.* Summary of uncertainties.

| | $CO_2$ (ppm) | $CH_4$ (ppb) | Method |
|---|---|---|---|
| Sensor repeatability | 0.05 | 5 | Combined standard uncertainty of SWS meas. in 3 min. |
| Reproducibility | <0.2 | <5 | SD of SWS measurements for one |

| | | | |
|---|---|---|---|
| (Time) | | week. | |
| Reproducibility (Sensor, Time) | 0.1[1] | -2±3 (min.) to 2±2 (max.) as annual mean  -10 (min.) to 5 (max.) as interval mean at every 10 ppb | Difference in NDIR (or SnO$_2$ sensor) values from CRDS values (Fig.6-7). |
| Scale uncertainty | 0.02 | 3.2 | See Section 2.4.2 and Supplement for CH$_4$. |

[1] Isotope effect in the NDIR. We plan to apply this correction to the NDIR measurements.